# LncRNA H19 Impairs Chemo and Radiotherapy in Tumorigenesis

**DOI:** 10.3390/ijms23158309

**Published:** 2022-07-27

**Authors:** Carlos Garcia-Padilla, Estefanía Lozano-Velasco, María del Mar Muñoz-Gallardo, Juan Manuel Castillo-Casas, Sheila Caño-Carrillo, Francisco José Martínez-Amaro, Virginio García-López, Amelia Aránega, Diego Franco, Virginio García-Martínez, Carmen López-Sánchez

**Affiliations:** 1Department of Human Anatomy and Embryology, Faculty of Medicine, Institute of Molecular Pathology Biomarkers, University of Extremadura, 06006 Badajoz, Spain; evelasco@ujaen.es (E.L.-V.); garcialopez@unex.es (V.G.-L.); virginio@unex.es (V.G.-M.); 2Department of Experimental Biology, University of Jaen, 23071 Jaen, Spain; mmmg0012@red.ujaen.es (M.d.M.M.-G.); jmcc0028@red.ujaen.es (J.M.C.-C.); scano@ujaen.es (S.C.-C.); fmamaro@ujaen.es (F.J.M.-A.); aaranega@ujaen.es (A.A.); dfranco@ujaen.es (D.F.); 3Fundación Medina, 18016 Granada, Spain

**Keywords:** lncRNA H19, chemo-resistance, radio-resistance, tumorigenesis

## Abstract

Various treatments based on drug administration and radiotherapy have been devoted to preventing, palliating, and defeating cancer, showing high efficiency against the progression of this disease. Recently, in this process, malignant cells have been found which are capable of triggering specific molecular mechanisms against current treatments, with negative consequences in the prognosis of the disease. It is therefore fundamental to understand the underlying mechanisms, including the genes—and their signaling pathway regulators—involved in the process, in order to fight tumor cells. Long non-coding RNAs, H19 in particular, have been revealed as powerful protective factors in various types of cancer. However, they have also evidenced their oncogenic role in multiple carcinomas, enhancing tumor cell proliferation, migration, and invasion. In this review, we analyze the role of lncRNA H19 impairing chemo and radiotherapy in tumorigenesis, including breast cancer, lung adenocarcinoma, glioma, and colorectal carcinoma.

## 1. Introduction

For decades, scientists have considered non-coding RNAs (ncRNAs) as a non-functional part of the genome, focusing their attention primarily on coding RNA biology. The sequencing of the human genome and later the ENCODE project have shown that more than 80% of the genome is transcribed in some type of RNA. Interestingly, only 3% of this transcribed genome corresponds to coding RNAs, suggesting that ncRNAs are as significant or more significant than coding RNAs [1,2]. It has been demonstrated that non-coding RNAs are essential for the regulation of cellular pathways and biological processes such as cell development, differentiation, growth, and homeostasis, as well as diseases [3,4,5,6].

According to their length, ncRNAs can be classified into: (i) small non-coding RNAs, with less than 200 nucleotides, including microRNAs, snoRNAs, piRNAs, and tRNAs [7]; and (ii) long non-coding RNAs (lncRNAs) with more than 200 nucleotides, including intronic lncRNAs, enhancer lncRNAs, circular lncRNAs, and intergenic lncRNAs [8].

With respect to microRNAs, these present 20–22 ribonucleotides in length on average and display the capacity to bind to the 3′ untranslated region (3′UTR) of coding RNAs by complementary base pairing, promoting their degradation and/or translational blockage. The role of microRNAs as post-transcriptional modulators has been widely described in multiple biological and cellular processes, including cell development, differentiation, growth, and homeostasis, as well as diseases [9,10,11,12].

In the case of lncRNAs, these are structurally similar to mRNAs since they are transcribed by RNA polymerase II, and have the same typical post-transcriptional modifications, in 5′ terminal cap and 3′ terminal poly (A) in particular. Notably, they lack the capacity to code proteins. Mechanistically, lncRNAs can act both as transcriptional regulators (by modulation of nuclear gene expression in different ways, including epigenetic landscape control, transcriptional complex scaffolding, and decoy molecules) and as post-transcriptional regulators (modulating microRNA degradation, mRNA stability, and/or protein translation). A deeper study of these lncRNAs roles will help us to better understand the regulation of multiple biological processes [13,14,15,16].

A particular lncRNA, H19 (produced by H19 gene), is abundantly expressed during embryonic development, mainly in derived tissues from the endoderm and mesoderm, and downregulated after birth, except for muscle tissues, such as skeletal and cardiac muscles [17,18,19]. H19 gene is highly conserved between primates and rodents, maintaining both structure and expression pattern in different species. However, it shows substantial differences in terms of the putative ORFs (open reading frames) arrangement present in this gene, which exhibits 5 exons separated by 4 introns, structurally [20]. While in humans a maximum of 13 different isoforms are observed, only 7 are identified in mice. H19 gene generates a 2.3 kb transcript with a cap in 5′ end and a polyA tail in 3′ end [21]. Subcellular expression of H19 depends on cellular type and biological context, showing its expression both in the cytoplasm and nucleus [22,23]. In the genome, H19 gene is located within a locus highly regulated by epigenomic machinery. This locus, named H19-Igf2, showing several differentially methylated regions (DMRs) and imprinting control regions (ICRs), leads to intensive modulation of genes contained in it, depending on the process and/or biological context [24,25]. Interestingly, H19 encodes the primary microRNA precursor for miR-675. The expression of this microRNA is not dependent on RNA H19 transcription, although both expressions are correlated in several biological processes [26,27]. The H19 gene has been studied for some time now, although its function is still not well defined. In particular, it acts as a trans-regulator of a group of co-expressed genes as part of an imprinted network, which is likely to control cellular homeostasis [24,25,26,27]

Taking into account that non-coding RNA genome has proved to play a pivotal role, both as tumor suppressor and oncogenic molecules in cancer therapy [28,29,30,31,32], it is critical to understand the underlying molecular mechanisms in favor of new therapeutic approaches for cancer treatment efficiency. Since the resistance to chemo and radiotherapy constitutes a crucial factor involved in disease relapse and metastasis [33,34], it is of great importance to establish the intrinsic factors involved in this process. Known molecular mechanisms of chemo and radioresistance include transporter pumps, oncogenes, tumor suppressor gene, mitochondrial alteration, DNA repair, autophagy, epithelial–mesenchymal transition (EMT), cancer stemness, and exosome and extensive epigenetic regulation, among others [35,36,37,38]. Noticeably, abnormal expression of lncRNA H19 has been found in different types of tumor cells, affecting cancer progression through different mechanisms, either as a suppressor or oncogenic gene, depending on cellular context [39,40,41,42,43,44,45]. For this reason, clinically, lncRNA H19 could be useful as a biomarker of diagnosis, therapy, and prognosis in cancer [46,47]. In this sense, those clear-cut mechanisms underlying the H19 regulatory roles in the biological progression of cancer requires further investigation.

Consequently, in this review, we analyze the impact of lncRNA H19 as a responsible factor in chemo and radioresistance of malignant cells, including their underpinning molecular mechanisms, specifically breast cancer, lung adenocarcinoma, glioma, and colorectal carcinoma.

## 2. LncRNA H19 Impairs Chemo and Radiotherapy in Breast Cancer

Breast cancer is the most frequently diagnosed malignancy and the second leading cause of cancer mortality in females worldwide [48]. Broadly, breast cancer patients can be classified into estrogen receptor positive (ER+)—corresponding to 70% of total cases—or estrogen receptor negative (ER−) [49,50]. Drug treatment against both types of breast cancer differs from each other, since several drugs are antagonists of estrogen receptors and exerting their functions blocking binding between ER and estrogens. This binding leads to disrupting the activation of downstream signaling pathway dependent of RE-estrogens interaction [51]. Currently, administration of three drugs, tamoxifen (TMX), doxorubicin (DOX), and paclitaxel (PTX), are considered as first-line of chemotherapy [52]. The application of any of these drugs separately or in combination with different types of drugs that enhance their cytotoxicity against malignant cells, considerably improves the clinical outcome of breast cancer, even though their continuous use results in acquired resistance of breast cancer cells against drug treatment [53,54,55]. To reverse such resistance and re-sensitize cells to different drugs, it is vitally important to understand the underlying molecular mechanisms that modulate their resistance acquisition. Several studies have pointed out the role of H19 as a powerful oncogenic agent in this process, pinpointing it as a critical modulator of chemoresistance (Figure 1).

Paclitaxel (PTX) is the most commonly used antitumor treatment drug on several types of carcinomas [55]. Mechanistically, PTX action includes several signaling pathways in which PTX modulates cellular processes that results in programmed cell death triggering. PTX is considered as the first-line treatment drug in breast cancer (BC), especially in triple negative breast cancer (TNBC) [56], since this subtype of carcinoma is not sensitive to estrogen receptor positive breast cancer cell. Unfortunately, the resistance of BC to PTX treatment is a great obstacle in clinical applications and one of the major causes of death associated with treatment failure. In breast cancer, PTX (Figure 1A) induces cellular apoptosis by interacting with β-tubulin and altering microtubules stability. PTX-resistance is acquired by mutations in α or β-tubulin, enhancing p53 and AKT activation signaling pathways or by dysregulation of apoptotic proteins [57,58]. Functional assays in both ER(+)-MCF-7 and ER(-)-Triple negative breast cancer (TNBC) carcinomas have shown that acquisition of resistance to PTX requires the upregulation of H19 (Figure 1B), which in turn blocks activation of several apoptotic pathways. Curiously, in ERα + breast cancer, H19 modulates resistance of PTX at both transcriptional and post-transcriptional levels. Si et al. (2016) [59] demonstrated that upregulation of H19 in MCF-7 cell line inhibited transcription of BCL-2 interacting killer protein (BIK)—a proapoptotic BH3-only member of the BCL-2 family which is prognostic for relapse and decreased overall survival of breast cancer [60]—by recruiting EZH2 subunit to the promoter of this gene, modulating H3 methylation at lysine 27. As result of BIK promoter methylation, PTX-resistance MCF-7 cells display increased proliferation rate and decreased cellular apoptosis [59]. Furthermore, H19 blocks miR-340-3p function—a known tumor suppressor miRNA involved in repression of EMT—acting as competitive sponge and avoiding degradation of tyrosine 3-monooxygenase/tryptophan 5-monoixygenase activation protein (YWHAZ), which in turn enhances activation of Wnt/β-catenin signaling pathway. As results of increased activity of YWHAZ-Wnt/β-catenin axis, PTX-resistance cells exhibit increased metastasis, invasion, and EMT. Additionally, the phenotype associated with H19 upregulation is recapitulated in both xenograft models and biopsies from patients with PTX resistance [61]. Unlike ER(+), increased H19 levels in TNBC promote activation of AKT by phosphorylation. As a consequence of enhancing AKT activity, BAX and cleaved caspase 3 are repressed, and cellular apoptosis is inhibited. Furthermore, downregulation of H19 in TNBC is translated into lower tumor growth rate in vivo and reduced cell proliferation accompanied by higher rates of apoptosis [62].

Doxorubicin (DOX) is a member of the anthracycline family and currently is one of the main treatments of choice for the treatment of both ER+ and ER- breast cancer [54]. DOX negatively affects the survival of malignant cells through different mechanisms of action (Figure 1A): (1) DOX is capable of mediating intercalation into DNA and disruption of topoisomesare-II function, avoiding DNA repair, and increasing cellular apoptosis; (2) DOX promotes generation of free radicals and their damage to cellular membranes, proteins, and DNA; (3) DOX deregulates several pivotal pathways involved in tumorogenesis such as PI3K/mTOR/AKT or ERK signaling [63]. Several studies have pointed out H19 as a major mediator of DOX chemoresistance (Figure 1C) by modulating MDR1/MDR4 and PARP1 expression [64,65]. Increased expression of H19 is required for the acquisition of DOX-resistance in several breast cancer cell lines. Curiously, downregulation of H19 reduces cell viability, lowers colony forming, and increases apoptosis under DOX treatment. Mechanistically, H19 promotes expression of Cullin 4A (CUL4A), a ubiquitin ligase component [66], which in turn enhances expression of ABCB1/4 genes that encoded MDR1/4 proteins, two members of the ATP-binding cassette family highly upregulated in several carcinomas, including breast cancer [67,68]. Both proteins exert pivotal functions in oncogenesis acting as inductors of multidrug resistance [69]. However, the molecular mechanisms dependent on MRD1/MDR4 are still unclear and require additional and intensive investigations. Wang et al. (2020) [65] showed that H19-induced DOX-chemoresistance is mediated by repression of PARP1. In MCF-7, high levels of H19 were correlated with downregulated PARP1 expression. Even though PARP1 is upregulated in several types of carcinomas and PARP1 inhibitors have been described as pivotal drugs against tumorigenesis, PARP1 has been shown to increase the antitumor activity of other drugs such as temozolomide and topotecan in preclinical studies, including models of pediatric cancers [70,71,72]. Curiously, it is capable of being packaged into exosomes from resistance cells and this diffusion towards sensitive cells leads to the acquisition of resistance against DOX treatment [73].

Tamoxifen (TMX), an anti-estrogen (Figure 1A), competitively inhibits estrogen binding to the ER and blocks the ER-mediated stimulation signal [53]. Five years of tamoxifen adjuvant therapy has been shown to safely reduce 15-year risk of breast cancer recurrence and death; however, a substantial group of patients was shown to eventually develop resistance (de novo or acquired) to tamoxifen. Although many molecular mechanisms of tamoxifen resistance have been described, including mutations in the ESR1 gene and the activation of alternative growth pathways, such as ERBB2/HER2, EGFR, IGF1R, and cyclin D1/CDK4/6 pathways, it remains necessary to gain an improved understanding of the potential mechanisms of tamoxifen resistance [74,75,76,77]. Several studies have pointed out the importance of upregulation of H19 in TMX resistance acquisition. Increased level of H19 in TMX-resistance cells (Figure 1D) is translated into enhanced autophagy activity and higher metabolism ratio by upregulation of Beclin1 (BECN1) and downregulation of N-acetyltransferase-1 (NAT1) transcription, respectively [78,79]. H19 downregulates the methylation state of Beclin1 promoter by binding and inhibiting S-adenosyl homocysteine hydrolase (SAHH). As consequence, DNMT3B function is reduced, resulting in lower ratio of methylation of Beclin1 promoter and thus higher Beclin1 expression [78]. Additionally, H19 modulate methylation of NAT1 promoter reducing transcription of it. NAT1 exert a pivotal role in metabolism of carcinogens and it is negatively correlated with a poor prognosis and aggressiveness of ER(+) breast cancer [79].

## 3. LncRNA H19 Impairs Chemo and Radiotherapy in Non-Small Cell Lung Cancer (NSCLC)

Epidermal growth factor receptor (EGFR) signaling is a receptor tyrosine kinase (RTK) mediated signaling commonly upregulated in many different tumors such as non-small-cell lung cancer, metastatic colorectal cancer, glioblastoma, pancreatic cancer, and breast cancer [80,81,82]. Upregulation of EGRF activity is caused by distinct mutations or truncations on both extracellular and/or kinase domain such as EGFRvIII truncations or L858R mutations, respectively. EGFR positively modulates PIK3/mTOR/AKT oncogenic signaling pathway by increasing phosphorylation, which in turn promotes proliferation, survival, and invasion of maligned cells [83]. Since the importance of it as a pivotal oncogenetic modulator, EGFR has been pinpointed as a powerful therapeutic target against different tumors types including small non cell lung cancer (NSCLC) [84,85]. Several inhibitors—such as Erlotinib and Gefinitib—(Figure 2) have been used to block EGFR function and activation of signaling pathway downstream PIK3/mTOR/AKT [86,87]. Although both drugs have been described as effective treatment against NSCLC, improving prognosis of disease and progression-free survival (PFS), many reports have revealed that continued administration is translated into acquired resistance. Underlying molecular mechanism of acquired resistance to chemotherapy is poor understanding. Chen et al. (2020) [88] demonstrated that acquisition of resistance to Erlotinib treatment in lung adenocarcinoma requires H19 silencing. Interestingly, both Erlotinib-resistant tumor from different patients and Erlotinib-resistant cell lines display low H19 expression levels. Functional assays demonstrated that gain-of-function of H19 in Erlotinib resistance cell lines result in restored drug sensibility. H19 reduces pyruvate kinase M1/2 (PKM2) protein level, promoting ubiquitin dependent degradation, which is essential for AKT phosphorylation and thus leads to disruption of PIK3/mTOR/AKT signaling. Furthermore, inhibition of PKM2 both in cells resistant to Erlotinib and in H19 knockout cells is translated into increased sensitivity to this drug indicating that Erlotinib-resistance is mediated by PKM2. Likewise, downregulation of H19 reduces sensibility to treatment with Erlotinib. Repression of H19 in Erlotinib-resistance increases PKM2 protein levels which in turn enhances AKT phosphorylation (Figure 2B). Curiously, AKT activity significantly reduces H19 expression, suggesting a negative feedback between H19-PKM2-AKT [88]. Conversely, Pan and Zhau (2020) [89] demonstrated that H19 is upregulated in exosomes of serum samples from non-responding erlotinib treatment patients suggesting that H19 could promote erlotinib resistance. Mechanically, H19 acts as a sponge of miR-615-3p, which in turn represses ATG7 translation, an oncogenic protein involved in autophagy response of different types of carcinomas [90,91]. As a consequence of H19-miR-615-3p competing mechanism, ATG7 translation is enhanced, leading to increased proliferation and survival of maligned cells (Figure 2B). Furthermore, H19 packaging into exosomes reduce Erlotinib sensibility to non-resistance Erlotinib cells pinpointing an oncogenic role of H19 [89]. The discrepancies between both studies suggest a complex role of H19 in the acquisition of resistance against Erlotinib that requires further study to be fully understood.

Resistance to Gefinitib is the major obstacle to improving the disease pattern of patients with advanced metastasis in NSCLC. Like Erlotinib, Gefinitib disrupts PI3K/mTOR/AKT pathway reducing phosphorylation. Particularly, Gefitinib removes a phosphate group from PIP3 active form, avoiding mTOR and AKT phosphorylation by increase of PIP2 intracellular pool (Figure 2A), which is indeed the inactive form [91]. Curiously, silencing of H19 expression enhances toxicity effects of Gefitinib in NSCLC. Furthermore, Gefinitib-resistant cell lines display high levels of H19, suggesting a pivotal role in acquisition of resistance. Functional assays have demonstrated that upregulation of H19 is accompanied by promotion of two oncogenic proteins, NFIB and DDHA1, that are involved in metastasis and angiogenesis, respectively. Zhou and Zhang (2020) [92] showed that co-treatment with Gefitinib and H19-shRNA drastically reduced NFIB expression by upregulation of PTEN and PDCD4. As result of downregulation of NFIB, which is considered as marker of metastasis in patients, invasion and spread out of maligned cell is reduced and blocking of progression of NSCLC is observed [93]. Likewise, upregulation of H19 is translated into increased levels of DDHA1 by competing with miR-148-3p (Figure 2C). H19-miR-148 binding avoids DDHA1 mRNA degradation and promote its translation, which in turn increases angiogenesis and is correlated with a poor patient diagnosis. Additionally, upregulation of miR-148 enhances the effects associated to H19 silencing in NSCLS while downregulation of this microRNA promotes the acquisition of Gefitinib resistance [92]. Curiously, H19 could be transfer via exosomes to non-resistance cells from maligned cell that display resistance to Gefinitib drug. As consequence of H19 transference, non-resistance cells do not respond to Gefitinib treatment worsening the prognosis of the disease and PFS by increasing both angiogenesis and metastasis of NSCLC [94].

Similar to observed in Gefitinib treatment, silencing of H19 enhanced sensitivity to radiotherapy by X-ray and carbons-ions in NSCLC cells. Zhao et al. (2021) [95] demonstrated that H19 was upregulated in radioresistant NSCLC (A549-R11) cells compared with control cells. Downregulation of H19 enhanced the sensitivity of NSCLC cell lines to X-ray and carbon ion irradiation. Functional assays have proven that H19 serves as a sponge of miR-130a-3p (Figure 2C), which downregulates WNK3 expression that in turn protects maligned cells from apoptosis [95].

Taking all the data described above into account and with the exception of the results obtained by Chen et al., 2020 [88], H19 displays an oncogenic role in lung adenocarcinoma, representing a powerful therapeutic target to avoid and counteract both radiotherapy and chemotherapy resistance.

## 4. LncRNA H19 Impairs Chemo and Radiotherapy in Glioma

Temozolomide (TMZ), an oral alkylating drug which delivers a methyl group to purine bases of DNA (O6-guanine; N7-guanine and N3-adenine), is frequently used together with radiotherapy as part of the first-line treatment of high-grade gliomas [96]. TMZ treatment (Figure 3) blocks the cell cycle at G2/M stage, leading to maligned cells towards cellular apoptosis [97]. Although TMZ treatment is effective and provides a significant improve in disease prognosis and patient survival, continuous use leads to acquired resistance. The underlying molecular mechanisms remain unclear but two cellular events are strongly connected to TMZ resistance: (1) extensive changes in epigenetic environment and (2) high generated oxidative stress in maligned cells [98]. Curiously, Duan et al. (2018) [99] proved that in response to rising oxidative stress, H19 is upregulated in glioma cells. Induced-H19 (Figure 3B) enhances activation of NF-κb signaling promoting expression of pivotal oncogenic genes such as Blc-2 and XIAP—which are upregulated in several human gliomas and to protect from apoptosis cellular [100,101]. As is the case for BCL-2 or XIAP, H19 increases Cyclin D1 expression, which mediates cycle cellular transition leading to a higher proliferative ratio and preventing malignant cells from being retained in the G2/M phase [102]. As a consequence of enhancing expression of those oncogenic genes, TMZ fails to induce cell cycle arrest and apoptosis resulting in acquisition of resistance against this drug [103]. Similar to observed with the genes dependent on NF-Kb pathway, upregulation of H19 positively mediates the expression of genes involved in multi-resistance against several drugs such as MDR1 or MRP. Additionally, H19 active beta-catenin signaling, which in turn enhances expression c-Myc and Survivin protecting glioma cells from programmed cell death. Furthermore, downregulation of cellular apoptosis levels is accompanied by increased spread out and invasion of maligned cells as result of enhanced EMT process [104]. H19 reduces expression of E-cadherin, a known protein involved in cell–cell junction, while increasing expression of Vimentin and ZEB1, inductor gene of the EMT process. Curiously, functional assays have demonstrated that TMZ treatment plus siRNA against H19 drastically reduce expression of genes described above. As a consequence of this joint administration, TMZ-glioma resistance is sensitive to effects of drug displaying a high ratio of cellular apoptosis and reduced EMT and metastasis. Therefore, combined use of both could contribute to improve clinical outcome of patients and better prognosis of disease.

Similar to resistance from TMZ treatment, H19 is upregulated by CREB1 protein under radiotherapy treatment against glioma. Functional assays have proven that H19 was involved in the cell cycle arrest, apoptosis, and DNA synthesis to modulate the radiation response of glioma cells and influenced their radioresistance [105].

## 5. LncRNA H19 Impairs Chemo and Radiotherapy in Colorectal Cancer

First evidence of involvement of H19 in chemoresistance of colorectal cancer cells was provided by Wu et al. (2017) [106]. Methotrexate (MTX) is a competitive inhibitor of dihydrofolate reductase (DHRF), a pivotal enzyme involved in intracellular folate metabolism [107]. DHFR is required to correct DNA synthesis and cellular growth. Alteration of DHRF function by MTX (Figure 4) results in increased programmed cell death and reduced cell proliferation and growth of maligned cells [108]. However, several carcinomas such as breast, bladder, head and neck cancers, osteogenic sarcoma, leukemia, and colorectal cancer display resistance against MTX. In colorectal cell lines, upregulation of H19 is translated into acquired resistance to MTX by upregulation of β-catenin signaling pathway (Figure 4B), which in turn activated expression of downstream transcriptional targets such as c-Myc, CCND1, CD44, and Oct3/4. Increased expression of genes dependent of β-catenin signaling pathway results in enhanced cellular proliferation and growth ratio [106].

Similar to observed in resistance against MTX, H19 also modulates resistance against oxaliplatin in colorectal cancer cells by activation of the β-catenin signaling pathway. Oxaliplatin (Figure 4A) disrupts DNA replication and transcription by forming intra-strand DNA adducts, but the downstream molecular events underlying the cytotoxic effects of this chemotherapeutic agent have not been well characterized [109]. Activation of β-catenin signaling pathway by H19 is mediated partially binding to miR-141-3p. H19 exerts as competitive sponge preventing miR-141-3p from targeting β-catenin mRNA and thus its degradation (Figure 4C). Curiously, resistance to oxaliplatin treatment is transferred by H19 contained in exosomes to sensitive cells by OXA-resistant cells both in vivo and in vitro colorectal carcinoma [110].

5-fluorouracil (5-FU) is a chemotherapeutical drug used to treat several carcinomas including colorectal cancer. 5-fluorouracil (Figure 4A) acts as an antimetabolite to reducing cell proliferation, by primarily blocking the enzyme thymidylate synthase and disrupting the thymidine formation necessary for DNA synthesis [111,112]. Resistance against 5-FU is dependent on upregulation of H19 (Figure 4D), which in turn reduces expression of retinoblastoma (RB) and p27Kip1-p23Kip1. As a consequence, SIRT1 expression is upregulated and autophagy procedure is activated [113].

## 6. Discussion and Future Perspectives

The role of lncRNAs in different aspects of carcinogenesis has demonstrated their key involvement in a dual role as promoters of tumorigenesis and tumor suppressor genes. In the last decade, the role of H19 has been widely described in many carcinomas, demonstrating a bivalent role in the appearance and diagnosis of the disease. The involvement of this lncRNA in the acquisition of resistance to both chemotherapy and radiotherapy has recently been demonstrated, suggesting its possible use as a therapeutic target. In fact, most treatment-resistant carcinomas displayed high H19 levels and H19 overexpression resulted in acquisition of resistance against the different treatments. Furthermore, silencing of this lncRNA makes resistant cells susceptible towards various drugs and ion radiation. Indeed, knockout of H19 in combination with different drug treatment enhanced anti-oncogenic response of malignant cell suggesting that H19 inhibitors could be of application together with current treatments to improve disease progression. Mechanistically, upregulation of H19 protects malignant cells from apoptosis by inducing expression of several antiapoptotic genes such as BCL-2 and multi-resistance drug cancer such as MDR1/2. Additionally, H19 increased autophagy response, which is positively correlated with several drug treatments. Unfortunately, malignant cells resistant to different treatments have the ability to convert surrounding cells into non-sensitive cells by emitting H19 containing exosomes. Taking these data into account, H19 could serve a powerful therapeutic target against several carcinomas to significantly improve prognosis and clinical outcome of patients.

The role of H19 in the acquisition of resistance to both chemotherapy and radiotherapy raises the question of the possibility that other lncRNAs or other elements of the non-coding genome could be exerting similar functions to those identified for H19 in functional gain-of-function assays. It would even be possible that the inhibition and/or overexpression of a certain set of lncRNAs could be considered a more effective treatment than those currently used. Therefore, a more in-depth knowledge on the role of the non-coding genome is critical to find new therapies that allow improving both the prognosis and the clinical outcome of several carcinomas.

## Figures and Tables

**Figure 1 ijms-23-08309-f001:**
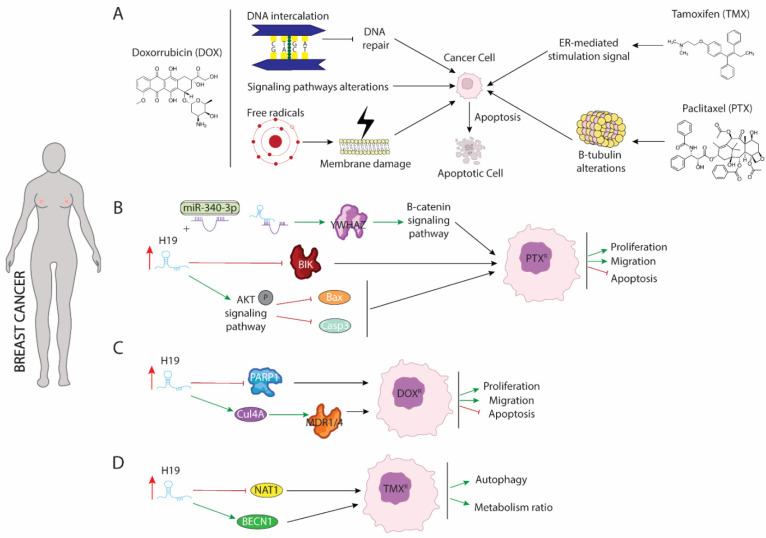
Schematic representation of molecular mechanism dependent of H19 function in breast cancer. (**A**) Modes of action of paclitaxel (PTX), doxorubicin (DOX), and tamoxifen (TMX) drugs involved in breast cancer treatment. (**B**) Role of H19 as a sponge to binding miR-340 enhancing expression of YWHAZ protein and increasing proliferation and migration while blocking apoptosis. (**C**) H19 increase expression of CUL4A and repress expression of PARP1 lead to DOX resistance in breast cancer. (**D**) H19 modulates methylation of BECN1 promotor and blocking NAT1 expression enhancing autophagy and metabolism procedures.

**Figure 2 ijms-23-08309-f002:**
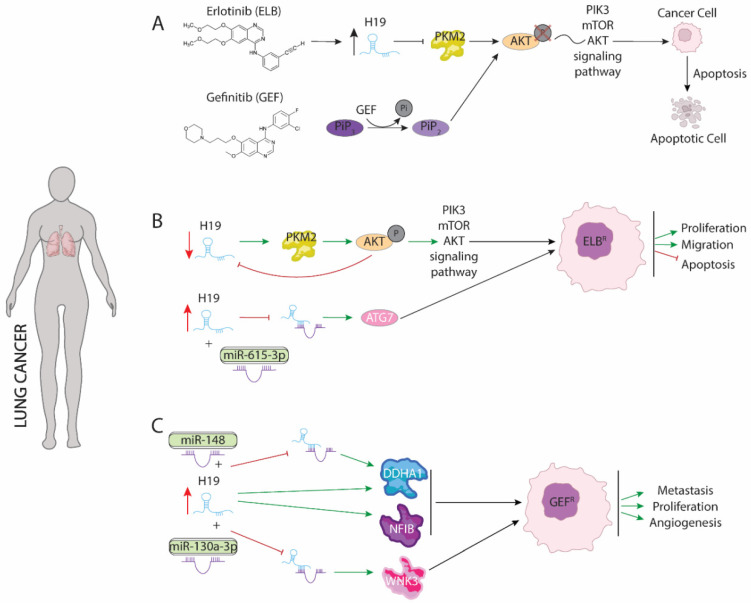
Schematic representation of molecular mechanism dependent of H19 function in lung cancer. (**A**) Modes of action of Erlotinib (ELB) and Gefinitib (GEF) drugs involved in lung cancer treatment. (**B**) Dual role of H19 in acquired resistance to Erlotinib modulating PKM2 and ATG7 proteins. (**C**) Role of H19 as a sponge to binding miR-148 and miR-130a enhancing expression of DDHA1, NFIB and WNK3 proteins increasing proliferation, metastasis, and angiogenesis procedures.

**Figure 3 ijms-23-08309-f003:**
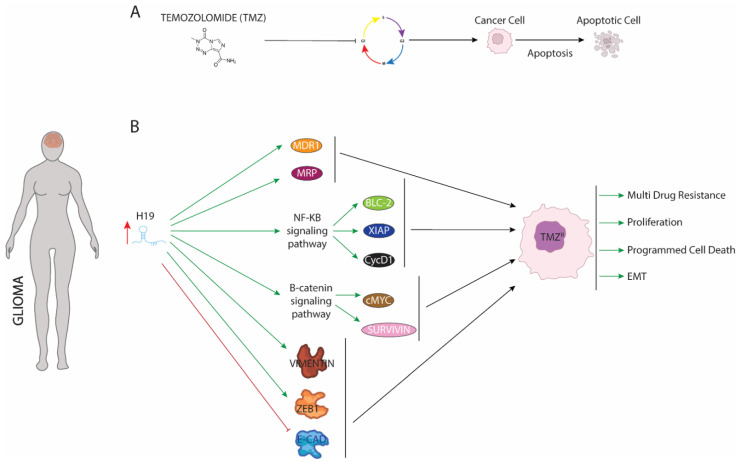
Schematic representation of molecular mechanism dependent of H19 function in glioma. (**A**) Mode of action of temozolomide. (**B**) Upregulation of H19 increase expression of proteins involved in multidrug resistance, apoptosis, and EMT procedures enhancing proliferation, programmed cell death, and EMT.

**Figure 4 ijms-23-08309-f004:**
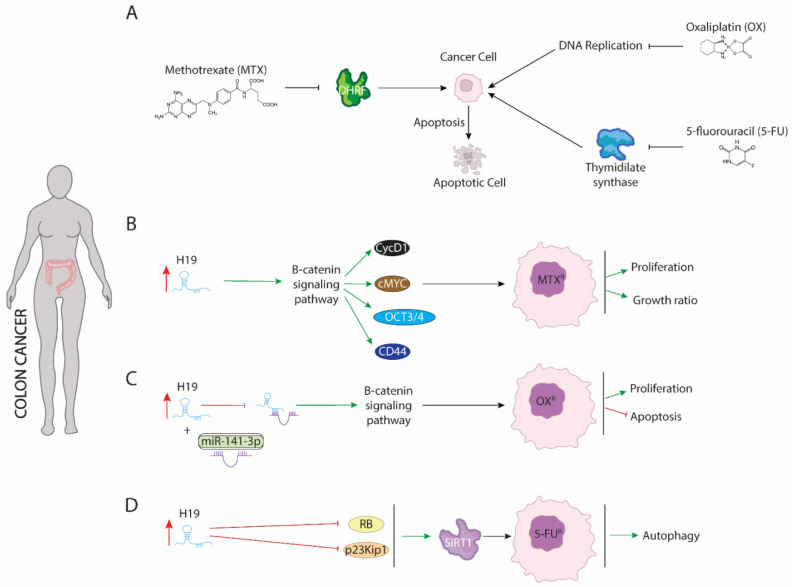
Schematic representation of molecular mechanism dependent of H19 function in colon cancer. (**A**) Mode action of methotrexate (MTX), oxaliplatin (OX), and 5-fluorouracil (5-FU) drugs involved in colon cancer treatment. (**B**) H19 modulate B-catenin signaling pathway genes such as CycD1 or cMYc increasing proliferation and growth ratio. (**C**) H19 acts as a sponge binding to miR-141 increasing expression B-catenin signaling pathway. (**D**) H19 blocking expression of RB and p23Kip1 leading to expression of SIRT1 which in turn, increases autophagy process.

## Data Availability

Not applicable.

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
