# Peer review of "LncRNA H19 Impairs Chemo and Radiotherapy in Tumorigenesis"

_ijms, 2022, doi:10.3390/ijms23158309_

Round 1
Reviewer 1 Report
In this manuscript, the authors reviewed the roles of lncRNA H19 in establishing chemo and radiotherapy resistance. They provided detailed explanations of the mechanism of action of the drugs commonly applied in treating breast cancer, lung adenocarcinoma, glioma, and colorectal carcinoma. They further discussed the underlying molecular impacts of the dysregulation of H19 on the process of cancer cells developing drug resistance. The review is well-structured and provides rich information to the authors interested in the field. I only have a few minor comments as follows:
Minor comments:
1. In line 67, please provide references when stating "Subcellular expression of H19 depends on cellular type and biological context, showing its expression both in the cytoplasm and nucleus."
2. In line 69, "differentially methylated regions (DHRs)" should use the acronyms of "DMRs".
3. In line 103-106, "oestrogen" should be replaced with "estrogen" (to keep it consistent across the whole manuscript). Also, "RE-oestrogens interaction" should be "ER-estrogen interaction".
4. In line 139, "suppressor tumour miRNA" should be "tumor suppressor miRNA".
5. In line 251, "Like to Erlotinib" should be "Like Erlotinib".
6. In line 384, "resistance-treatment carcinomas" should be "treatment-resistant carcinomas".
7. Throughout the manuscript there are too many attribute clauses, making the sentences too long to comprehend. Try to break the long sentences into shorter sentences, if possible.
Author Response
First of all, we would like to thank the reviewer for their constructive comments that certainly will be helpful in this manuscript.
In this manuscript, the authors reviewed the roles of lncRNA H19 in establishing chemo and radiotherapy resistance. They provided detailed explanations of the mechanism of action of the drugs commonly applied in treating breast cancer, lung adenocarcinoma, glioma, and colorectal carcinoma. They further discussed the underlying molecular impacts of the dysregulation of H19 on the process of cancer cells developing drug resistance. The review is well-structured and provides rich information to the authors interested in the field. I only have a few minor comments as follows:
Minor points to be addressed
Minor comments:
- In line 67, please provide references when stating "Subcellular expression of H19 depends on cellular type and biological context, showing its expression both in the cytoplasm and nucleus." References added
- In line 69, "differentially methylated regions (DHRs)" should use the acronyms of "DMRs". Changed added
- In line 103-106, "oestrogen" should be replaced with "estrogen" (to keep it consistent across the whole manuscript). Also, "RE-oestrogens interaction" should be "ER-estrogen interaction". Changed added
- In line 139, "suppressor tumour miRNA" should be "tumor suppressor miRNA". Changed added
- In line 251, "Like to Erlotinib" should be "Like Erlotinib". Changed added
- In line 384, "resistance-treatment carcinomas" should be "treatment-resistant carcinomas". Changed added
- Throughout the manuscript there are too many attribute clauses, making the sentences too long to comprehend. Try to break the long sentences into shorter sentences, if possible. Following the reviewer´s recommendation sentences have been reformulated to make them shorter (see lines 103-105, 108-109 or 308-309) For example:
Original sentence: Since several drugs are antagonists of estrogen receptors and exerting their functions blocking binding between ER and estrogens, lead to disrupting the activation of downstream signalling pathway dependent of RE-estrogens interaction
Current sentence: Since several drugs are antagonists of estrogen receptors and exerting their functions blocking binding between ER and estrogens. This binding lead to disrupting the activation of downstream signalling pathway dependent of RE-estrogens interaction

Reviewer 2 Report
The review manuscript “LncRNA H19 impairs chemo and radiotherapy in tumorigenesis” by Garcia-Padilla et al., well summarized how lncRNA H19 RNAs contribute to chemotherapy and radiotherapy and it serves as a potential therapeutic target against diverse cancer types. Overall the manuscript was well written and cited important publications in this field. I have just one minor comment about references. Authors should double check reference format and remove [CrossRef], [PubMed], PMID, PMCIP, Epub, etc. Other than that, I don’t have any further comments. Thank you.
Author Response
Reply to reviewer #2
First of all, we would like to thank the reviewer for their constructive comments that certainly will be helpful in this manuscript.
The review manuscript “LncRNA H19 impairs chemo and radiotherapy in tumorigenesis” by Garcia-Padilla et al., well summarized how lncRNA H19 RNAs contribute to chemotherapy and radiotherapy and it serves as a potential therapeutic target against diverse cancer types. Overall the manuscript was well written and cited important publications in this field.
Minor points to be addressed
I have just one minor comment about references. Authors should double check reference format and remove [CrossRef], [PubMed], PMID, PMCIP, Epub, etc. Other than that, I don’t have any further comments. Thank you.
Following the reviewer´s recommendation the references have been revised and remove [CrossRef], [PubMed], PMID, PMCIP, Epub from text.
